# Using Rare Earth Elements (REE) to Decipher the Origin of Ore Fluids Associated with Granite Intrusions

**Xue-Ming Yang**

Manitoba Geological Survey, 360-1395 Ellice Avenue, Winnipeg, MB R3G 3P2, Canada; eric.yang@gov.mb.ca

**Abstract:** A practical method is presented to estimate rare earth element (REE) concentrations in a magmatic fluid phase in equilibrium with water-saturated granitic melts based on empirical fluid–melt partition coefficients of REE ($k_P^{REE}$). The values of $k_P^{REE}$ can be calculated from a set of new polynomial equations linking to the Cl molality ($m_{Cl}^v$) of the magmatic fluid phase associated with granitic melts, which are established via a statistical analysis of the existing experimental dataset. These equations may be applied to the entire pressure range (0.1 to 10.0 kb) within the continental crust. Also, the results indicate that light REEs (LREE) behave differently in magmatic fluids, i.e., either being fluid compatible with higher $m_{Cl}^v$ or fluid incompatible with lower $m_{Cl}^v$ values. In contrast, heavy REEs (HREE) are exclusively fluid incompatible, and partition favorably into granitic melts. Consequently, magmatic fluids tend to be rich in LREE relative to HREE, leading to REE fractionation during the evolution of magmatic hydrothermal systems. The maximum $k_P^{REE}$ value for each element is predicted and presented in a REE distribution diagram constrained by the threshold value of $m_{Cl}^v$. The REE contents of the granitic melt are approximated by whole-rock analysis, so that REE concentrations in the associated magmatic fluid phase would be estimated from the value of $k_P^{REE}$ given chemical equilibrium. Two examples are provided, which show the use of this method as a REE tracer to fingerprint the source of ore fluids responsible for the Lake George intrusion-related Au–Sb deposit in New Brunswick (Canada), and the Bakircay Cu–Au (–Mo) porphyry systems in northern Turkey.

**Keywords:** REE distribution pattern; REE fluid–melt partition coefficient; granite; intrusion-related gold system; porphyry copper (gold) system

## 1. Introduction

Using rare earth elements (REE) to study the origin of ore fluids associated with granite (*sensu lato*) intrusions has been made possible and attractive, thanks to a number of high pressure (P)-temperature (T) experiments on REE partitioning between magmatic fluid phases and granitic melts [1–3]. This technique has been proven to be reliable and cost-effective in probing the sources of ore fluids associated with granite intrusions [4–6].

Experimental studies indicate that the REE fluid–melt partition coefficient ($k_P^{REE}$) (see Table 1 for explanations of symbols) is dominantly dependent upon fluid composition (i.e., Cl molality $m_{Cl}^v$), and is also controlled to some extent by melt composition (X) [which can be described by the Al saturation index (ASI = molar Al/(Na + K + Ca)] and pressure (1.25 to 10.0 kb), but is not notably affected by temperature (750 to 950 °C) [3]. In a strongly peraluminous (ASI >1.1) granitic melt system associated with a fluid phase with a large range of $m_{Cl}^v$ from 0.1 to 6.0 *M*, the values of $k_P^{La}$ range from ca. 0.035 to 0.150 [3]. In contrast, in a moderately peraluminous (ASI = 1.0 to 1.1) to metaluminous (ASI <1.0) melt system associated with a fluid phase with the same $m_{Cl}^v$ range, $k_P^{La}$ values range from ca. 0.005 to 1.5 [2,3]. These experimental observations suggest that $k_P^{La}$ values could be estimated over a range of X-P conditions.

**Table 1.** Symbols used in this paper. REE: rare earth elements.

| Symbol | Explanation |
|---|---|
| ASI | Aluminum saturation index that is expressed as molar ratio of Al/(Ca + Na + K) of granitic melts (or granites) |
| $C_{Cl}^m$ | Cl content (wt %) of granitic melt |
| $C_{REE}^m$ | REE concentration of granitic melt, e.g., $C_{La}^m$ stands for La content (ppm) in the melt |
| $C_{REE}^v$ | REE concentration of magmatic fluid (or magmatic fluid phase) |
| $k_P^{Cl}$ | Cl partition coefficient between magmatic fluid and granitic melt, which is defined by the ratio of $m_{Cl}^v / m_{Cl}^m$ |
| $k_P^{REE}$ | REE partition coefficient between magmatic fluid and granite melt; for example, $k_P^{La}$ denotes La partition coefficient defined by $C_{La}^v / C_{La}^m$ ratio, and so on |
| $m_{Cl}^m$ | Cl molality of granitic melt |
| $m_{Cl}^v$ | Cl molality of magmatic fluid (i.e., aqueous fluid phase) |

It is well known that granitic melts could lose a magmatic vapor phase, as evidenced in trapped fluid inclusions in primary quartz phenocrysts [6–9], to surrounding country rocks during cooling and decompression as well as crystallization-induced degassing (second boiling), resulting in hydrothermal alteration and sometimes mineralization, e.g., porphyry Cu–Au (–Mo) [4], intrusion-related Au [5,7,10], Sn deposits related to rare metal granites [11–14], scheelite skarn [15], and W–Au-bearing quartz veins [16]. As much as 5 wt % water containing various metals and/or ligands may be lost to the surrounding country rocks upon the cooling of granite magmas during their ascent and emplacement [17]. This may be concentrated to form an ore deposit if the geological settings are favorable. The composition of the magmatic fluid phase can be modeled by studying ore-forming elements (e.g., Cu, Mo) partitioning between the magmatic fluid phase and granitic melts through high P–T experimental and empirical investigations [18–20]. Therefore, it is possible to determine the roles played by the magmatic fluids, which are associated with granitic magmas, during mineralization by comparing the geochemical data (e.g., REE) of the ore deposits and associated alteration with the composition of the modeled magmatic fluids based on the fluid–granitic melt partitioning data.

This paper presents a practical method to calculate REE concentrations in the magmatic fluid phase based on a set of new polynomial equations that can be used for the estimation of REE fluid–granitic melt partition coefficients over a wide range of P-T-X conditions. This technique is then used for tracking the sources of ore fluids that are responsible for the formation of two typical mineral deposits: (1) the Lake George intrusion-related Sb–Au–W–Mo deposit (NB, Canada) characterized by low to medium salinity ore fluids; and (2) the Bakircay Cu–Au (–Mo) porphyry system in northern Turkey, which may have been formed by ore fluids with medium to high salinities.

## 2. Methodology

An element partition coefficient between a magmatic fluid phase and a granitic melt is defined by the ratio of its concentration in the magmatic fluid phase to that in the melt [1,21] (Table 1). For instance, the REE partition coefficient $k_P^{REE}$ between the magmatic fluid phase and silicate melt can be expressed as the ratio of $C_{REE}^v/C_{REE}^m$; similarly, the Cl partition coefficient $k_P^{Cl}$ is defined as the ratio of $m_{Cl}^v/m_{Cl}^m$ (Table 1). The partition coefficient is dimensionless, and is controlled by various variables, such as the pressure, temperature, and composition of magmatic hydrothermal systems [1–3,5,17–23]. It is also known that the value of $k_P^{Eu}$ could be affected by the oxygen fugacity ($f$O$_2$), as Eu has two valences (i.e., Eu$^{3+}$, Eu$^{2+}$) that depend upon $f$O$_2$ [1,3,10].

Estimates of REE concentration in the magmatic fluid will be carried out by using empirical values of $k_P^{REE}$ obtained by the new equations presented in this study (see below), REE bulk-rock data from fresh granite samples, which are assumed to approximate that of the granitic melt in equilibrium with

the magmatic fluid, and altered rock, which resulted from the interaction or reaction of the fresh rock with the magmatic fluid.

## 2.1. Relationship of REE Fluid–Granitic Melt Partition Coefficient to Cl Molality

Calculation of REE partition coefficients ($k_P^{REE}$) between a magmatic fluid and granite melt has been presented previously by [5] according to experimental results from [1,2]. Briefly, it can be described by a linear function of cubic power in terms of $m_{Cl}^v$, but the Eu fluid–granitic melt partition coefficient ($k_P^{Eu}$) is related to the fifth power of $m_{Cl}^v$ when the aqueous fluids have relatively low values of $m_{Cl}^v$. This confirms the experimental observations by [1,2]. The mathematical relationship of $k_P^{REE}$ to the low range of $m_{Cl}^v$ for aqueous fluids is provided on the basis of a least square regression analysis of the experimental dataset. However, such a linear relationship cannot extend to higher values of $m_{Cl}^v$ (e.g., >3.5 M), which was also noticed by [3]. Therefore, it is necessary to re-assess the existing experimental data presented in [1–3]. Here, these datasets (Supplementary Table S1) are plotted on Figure 1, using La to represent LREE and Yb to represent HREE (note: the other REEs are not shown). To best fit the entire dataset, a set of new polynomial equations linking $k_P^{REE}$ with $m_{Cl}^v$ can be obtained for each REE, as shown in Table 2. However, as evident in Figure 1, more experimental data are required to fill the data gap between the two extremes. Remarkably, these equations display a relatively higher coefficient of determination ($R^2$ ranging from 0.943 to 0.969, i.e., the square of the correlation coefficient, R; see Table 2) when compared with those ($R^2$ ranging 0.90 to 0.95) presented by [5]. Based on the equations presented in Table 2 respectively for La and Yb, $k_P^{La}$ can be readily calculated, for example, for a magmatic fluid with an $m_{Cl}^v$ value of 1.0 M, to be 0.054, and $k_P^{Yb}$ to be 0.024.

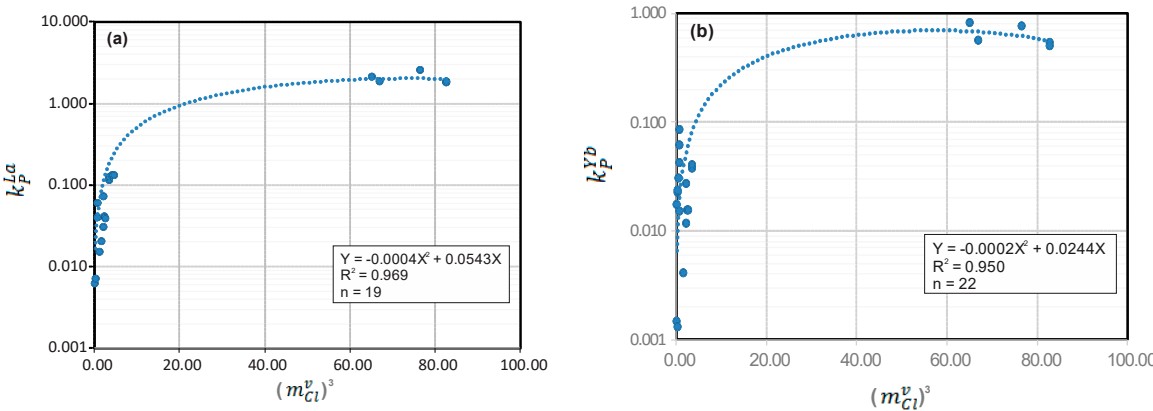

**Figure 1.** Plot of REE fluid–granitic melt partition coefficient ($k_P^{REE}$) versus Cl molality ($m_{Cl}^v$) of magmatic fluid phase: (**a**) $k_P^{La}$ versus $(m_{Cl}^v)^3$, and (**b**) $k_P^{Yb}$ versus $(m_{Cl}^v)^3$. Data used for this plot are from [1–3] (Supplementary Table S1). In each of the equations, Y denotes $k_P^{REE}$; X represents $(m_{Cl}^v)^3$, except for Eu, where $e$ is $(m_{Cl}^v)^5$ (Table 2); $R^2$ is the coefficient of determination (that is, the square of the correlation coefficient, $R$); and n is the number of experimental points. It is worth noting that the uncertainties of such an estimation for $k_P^{REE}$, using the equations listed in Table 2, are not well constrained, although they must have been within the errors of the original dataset [1–3], i.e., ±0.025 for light rare earth elements (LREE), and ±0.030 for heavy rare earth elements (HREE). (**a**) Plot of $k_P^{La}$ vs. $(m_{Cl}^v)^3$, (**b**) plot of $k_P^{Yb}$ vs. $(m_{Cl}^v)^3$.

The REE fluid–granitic melt partition coefficients ($k_P^{REE}$) data used for the statistical analysis of this study (Supplementary Table S1) were acquired under experimental conditions of 4.0 kb and 800 °C buffered to quartz–fayalite–magnetite by [1], at 2.0 kb and 800 °C by [2], and at 0.2 to 10 kb and 750 to 950 °C by [3]. Numerical analysis of these datasets strongly suggests that $k_P^{REE}$ is dominantly controlled by the Cl molality ($m_{Cl}^v$) of the fluids in equilibrium with the granitic melts, which is consistent with the experimental observations by [1–3]. The experimental results indicate that the $k_P^{REE}$ values are not influenced by temperature, although the pressure effect appears to be notable, i.e., the values of $k_P^{REE}$

for trivalent REEs would increase with decreasing pressure. Interestingly, [3] have pointed out that the pressure (2.0 to 9.0 kb) effect on $k_P^{La}$ values is not discernible for peraluminous granitic melts, although such an effect is slightly manifest for metaluminous melts. For example, the value of $k_P^{La}$ increases from 0.03 at 3.0 kb to 0.12 at 7.0 kb when a fluid with an $m_{Cl}^v$ of 6.0 $M$ is associated with a metaluminous granitic melt [3]. Obviously, more high P–T experiments are required to cover a range of pressures from the extreme lower to extreme upper continental crust. Furthermore, it has been well constrained that only $k_P^{Cl}$ is pressure dependent [17]. Consequently, the value of $m_{Cl}^v$ in magmatic fluids is only related to pressure, which is the main controlling factor on the values of $k_P^{REE}$ (Table 2). Therefore, these equations (Table 2) may be used to calculate $k_P^{REE}$ values for a wide range of pressure (0.1 to 10.0 kb) within the continental crust, and to estimate REE concentrations in magmatic fluids, although the limitation of such a practical estimation should be kept in mind.

**Table 2.** Relationship of REE fluid–granite melt partition coefficients ($k_P^{REE}$) to Cl molality ($m_{Cl}^v$) of magmatic fluids.

| REE | Equation for Estimating $k_P^{REE}$ | $R^2$ | Number of the Experimental Data Used for Polynomial Analysis (*n*) |
|---|---|---|---|
| La | $k_p^{La} = -0.0004x^2 + 0.0543x$ | 0.969 | 19 |
| Ce | $k_p^{Ce} = -0.0005x^2 + 0.0649x$ | 0.964 | 20 |
| Nd | $k_p^{Nd} = -0.0004x^2 + 0.0588x$ | 0.958 | 13 |
| Sm | $k_p^{Sm} = -0.0004x^2 + 0.0519x$ | 0.950 | 13 |
| Eu | $k_p^{Eu} = -0.0000007e^2 + 0.0017e$ | 0.956 | 20 |
| Gd | $k_p^{Gd} = -0.0003x^2 + 0.0419x$ | 0.943 | 22 |
| Tb | $k_p^{Tb} = -0.0003x^2 + 0.0413x$ | 0.956 | 13 |
| Ho | $k_p^{Ho} = -0.0003x^2 + 0.0338x$ | 0.965 | 11 |
| Yb | $k_p^{Yb} = -0.0002x^2 + 0.0244x$ | 0.950 | 22 |
| Lu | $k_p^{Lu} = -0.0002x^2 + 0.0207x$ | 0.962 | 13 |

Note: $x$ denotes $(m_{Cl}^v)^3$ in the equations except for Eu, in which instead of $x$, a different variable $e$ is used that is equal to $(m_{Cl}^v)^5$.

A virtual examination of Figure 1 suggests that the value of $k_P^{REE}$ would approach its maximum with increasing $(m_{Cl}^v)^3$ to a certain value, or a threshold value. When $(m_{Cl}^v)^3$ is much smaller than this threshold value, $k_P^{REE}$ has a linear relationship to the variable as described by [1,2,5]. As long as the $m_{Cl}^v$ value of a magmatic fluid reaches the threshold value, the value of $k_P^{REE}$ would reach the maximum, and then appear to remain constant regardless of how much the concentration of Cl in the fluid is increased. This conclusion is also achieved by a numerical analysis (i.e., differentiation) of the equations listed in Table 2. This practice is able to predict the $(m_{Cl}^v)^3$ threshold value and maximum $k_P^{REE}$ value for each individual element (Table 3). Obviously, the implication of such threshold values and/or parameters needs to be explored further, and tested by more experimental work, because the existing dataset used in this paper (Supplementary Table S1) is obviously limited. Despite the limitations of the data, the functions presented in Table 2 appear to be applicable over the entire pressure range within the continental crust (e.g., 0.1 to 10.0 kb) owing to the maximum $k_P^{REE}$ values relying only upon the Cl molality threshold values $(m_{Cl}^v)^3$ of the magmatic fluids (Figure 2; Table 3) irrespective of the pressure on the fluid–melt systems in question. Hence, it is recommended that the maximum value of $k_P^{REE}$ for each REE be used to estimate the REE contents of the magmatic fluid once the Cl molality of the fluid reaches the threshold value.

**Table 3.** Predicted Cl molality threshold values $(m_{Cl}^v)^3$ of magmatic fluids (note: for Eu, it should be $(m_{Cl}^v)^5$) and maximum values of REE fluid–granite melt partition coefficients of $k_P^{REE}$ from this study.

| REE | Threshold Value $(m_{Cl}^v)^3$ of Fluid (Except for Eu That is $(m_{Cl}^v)^5$) | $m_{Cl}^v$ of Fluid (M) | Maximum Value of $k_P^{REE}$ |
|---|---|---|---|
| La | 67.88 | 4.08 | 1.843 |
| Ce | 64.90 | 4.02 | 2.106 |
| Nd | 73.50 | 4.19 | 2.161 |
| Sm | 64.88 | 4.02 | 1.684 |
| Eu | 1214.29 | 4.14 | 1.961 |
| Gd | 69.83 | 4.12 | 1.463 |
| Tb | 68.83 | 4.10 | 1.421 |
| Ho | 56.33 | 3.83 | 0.952 |
| Yb | 61.00 | 3.94 | 0.744 |
| Lu | 51.75 | 3.73 | 0.536 |

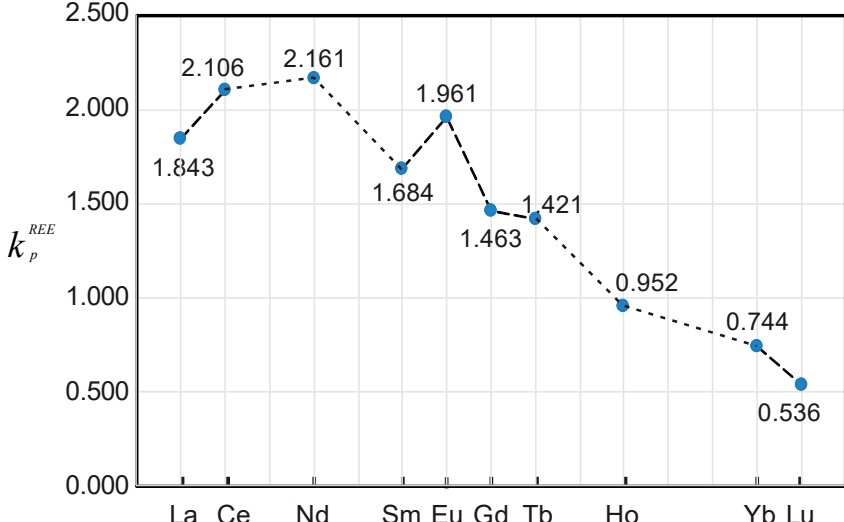

**Figure 2.** Plot of the predicted maximum values of REE partition coefficients ($k_P^{REE}$) between magmatic fluids and granitic melts based on the equations in Table 2. Number marked on the graph is the maximum value of predicted $k_P^{REE}$ for each element, e.g., La is equal to 1.843.

Figure 2 indicates that REE partitioning between magmatic fluids and granite melts must result in LREE enrichment relative to HREE in the fluids, which is consistent with the observations of the alteration halos associated with the Cu–Au (–Mo) porphyry systems [4], intrusion-related Au systems [5,10], and with regard to the REE content of natural fluid inclusions [6]. Also, it is likely that the fluids with relatively low $m_{Cl}^v$ values would display a negative Eu anomaly, as indicated by a numerical analysis of the relationship of $k_P^{REE}$ to $m_{Cl}^v$ (Table 2), unless their $m_{Cl}^v$ values are close to the threshold value that would result in a positive Eu anomaly (Figure 2), depending upon the REE content of the related granite melts. However, this effect could be influenced by the value of $m_{Cl}^v$, which is a function of pressure [17,21–24]. Maximum partition coefficients for other REE (e.g., Pr, Dy, Er, Tm) may be obtained by extrapolation from the data presented in Figure 2. Further extrapolation is not encouraged, although such a practice would provide a basis for future work to determine the values of $k_P^{REE}$ to $m_{Cl}^v$ experimentally.

Furthermore, the equations (Table 2) can also be used for the evaluation of REE behavior in a fluid–granitic melt system. When the value of $k_P^{REE}$ is $\geq$ 1.0, REE is fluid-compatible and prefers partitioning into the magmatic fluid. Since REE is granitic melt-compatible when $k_P^{REE} < 1.0$, light rare earth element (LREE) (e.g., La) would not become fluid compatible until the value of $(m_{Cl}^v)^3$ is higher than 21.97 (i.e., $m_{Cl}^v \geq 2.80$ M), which is obtained by solving the equation when $k_P^{La}$ equals to 1.0

(see Figures 1a and 3). In other words, La would be fluid incompatible and favor the granitic melt if the fluid has the $m_{Cl}^v$ value below 2.80 *M*. Therefore, LREE behaves differently in the fluid–granite melt system, depending upon the value of $m_{Cl}^v$ for the magmatic fluids. Interestingly, Eu requires a much higher $m_{Cl}^v$ value to become fluid compatible than the other LREE, indicating that it is melt compatible in equilibrium with magmatic fluids with low $m_{Cl}^v$, and therefore, such fluids commonly have a negative Eu anomaly. If magmatic fluids reach the Cl molality threshold value $(m_{Cl}^v)^3$ (Table 3; Figure 2), they would display a positive Eu anomaly. However, heavy rare earth element (HREE, e.g., Yb), is typically fluid incompatible, and prefers partitioning into the granitic melt (Figure 1b). The maximum value of $k_P^{Yb}$ is 0.744 when $(m_{Cl}^v)^3$ is equal to its threshold value of 61.00 (or $m_{Cl}^v = 3.94$ *M*) for fluids associated with granitic melts, leading to REE fractionation. Here, LREE are enriched relative to HREE during evolution of the magmatic hydrothermal systems.

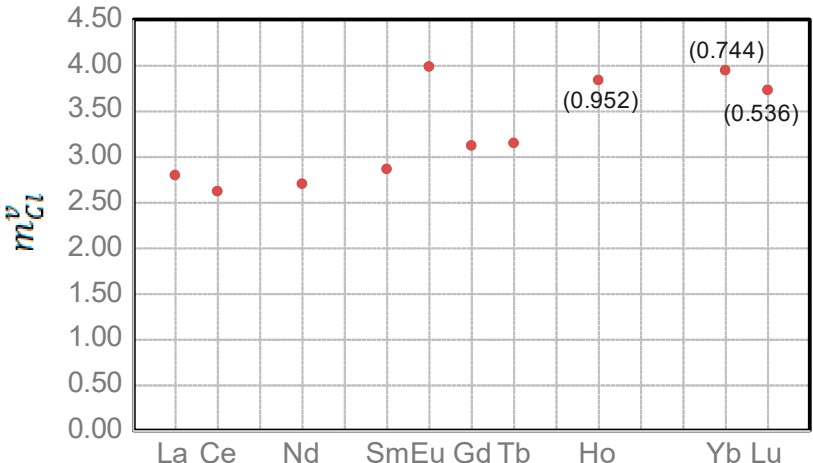

**Figure 3.** Diagram of $m_{Cl}^v$ values required for $k_P^{REE} \geq 1.0$ so that LREEs become fluid compatible. Note that HREEs are exclusively fluid incompatible, partitioning into granitic melts because their $k_P^{REE}$ are always <1.0 and would reach maximum values (number in brackets; Table 3) when $(m_{Cl}^v)^3$ achieve their respective threshold values.

More recently, experimental work [25] showed that REEs prefer partitioning into carbonate melt with fluid–melt partition coefficients <1.0 at 1–2 kb and 700–800 °C, but the associated fluids are relatively enriched in HREE compared to LREE. However, this experimental work suggests that REE fractionation in the fluid–carbonatite melt system distinctly differs from that in the fluid–granite melt system, as discussed in this study.

*2.2. Calculation of Cl Molality ($m_{Cl}^v$)*

As pointed out by [24], it is not easy to obtain the Cl concentration in a granitic melt. To make it simple, the Cl molality $m_{Cl}^m$ (unit in M or moles/kg) of a granitic melt can be calculated by the Cl content of the granite, as indicated by the equation below.

$$m_{Cl}^m = \left(C_{Cl}^m \div 35.5\right) \times \left[\left(100 - C_{Cl}^m\right) \div 1000\right] \tag{1}$$

where the Cl content ($C_{Cl}^m$) in the granitic melt is approximated by the analysis of the whole-rock sample with its unit in wt %.

Such an exercise of using Equation (1) to calculate the Cl molality ($m_{Cl}^v$) for a granite sample would provide a minimum value for the corresponding granitic melt, because a large amount of Cl may have been lost from the melt during cooling [17,23]. This is unavoidable because of the high Cl partition coefficient between the magmatic fluid phase and granitic melt under magmatic conditions [21,22,24]. Table 4 shows the results of this calculation by using two examples.

**Table 4.** Calculation of REE concentrations in a magmatic fluid.

| | Lake Geoge Granodiorite [5] | | Bakircay Granodiorite [4] | |
|---|---|---|---|---|
| | A | B | D | E |
| La (ppm) | 30.9 | 30.9 | 12.7 | 12.7 |
| Ce | 51.2 | 51.2 | 25.8 | 25.8 |
| Nd | 13.7 | 13.7 | 12.1 | 12.1 |
| Sm | 4.2 | 4.2 | 2.17 | 2.17 |
| Eu | 1.08 | 1.08 | 0.54 | 0.54 |
| Gd | 4.8 | 4.8 | 2.24 | 2.24 |
| Tb | 0.8 | 0.8 | | |
| Dy | | | 2.25 | 2.25 |
| Er | | | 1.16 | 1.16 |
| Yb | 2.2 | 2.2 | 1.1 | 1.1 |
| Lu | 0.3 | 0.3 | | |
| Cl | 1000 | 15 wt % NaCl$_{eqv.}$ in fluid inclusion [17] | 17 wt % NaCl$_{eqv.}$ in primary inclusion | 52% NaCl$_{eqv.}$ Fluid inclusion, 18.52 $M$ >Threshold |
| **Calculation of REE Fluid-Granitic Melt Partition Coefficient** | | | | **Equation** |
| $m^m_{Cl}$ | 0.028 | | | | (1) |
| $m^v_{Cl}$ | 1.23 | 3.02 | 3.50 | >Threshold | (2) |
| $k^{Cl}_P$ | 43.5 | | | | at 2 kb [17] |
| $k^{La}_P$ | 0.099 | 1.189 | 1.594 | 1.843 | from Table 2 |
| $k^{Ce}_P$ | 0.118 | 1.405 | 1.864 | 2.106 | from Table 2 |
| $k^{Nd}_P$ | 0.107 | 1.313 | 1.787 | 2.161 | from Table 2 |
| $k^{Sm}_P$ | 0.094 | 1.123 | 1.491 | 1.684 | from Table 2 |
| $k^{Eu}_P$ | 0.005 | 0.381 | 0.701 | 1.961 | from Table 2 |
| $k^{Gd}_P$ | 0.076 | 0.924 | 1.246 | 1.463 | from Table 2 |
| $k^{Tb}_P$ | 0.075 | 0.908 | 1.220 | 1.421 | from Table 2 |
| $k^{Ho}_P$ | 0.061 | 0.702 | 0.898 | 0.952 | from Table 2 |
| $k^{Yb}_P$ | 0.044 | 0.519 | 0.679 | 0.744 | from Table 2 |
| $k^{Lu}_P$ | 0.038 | 0.418 | 0.520 | 0.536 | from Table 2 |
| **Calculation of REE Concentrations (ppm) in MVPs** | | | | |
| $c^v_{La}$ | 3.05 | 36.74 | 20.24 | 23.41 |
| $c^v_{Ce}$ | 6.04 | 71.92 | 48.10 | 54.33 |
| $c^v_{Nd}$ | 1.47 | 17.98 | 21.62 | 26.15 |
| $c^v_{Sm}$ | 0.40 | 4.72 | 3.23 | 3.65 |
| $c^v_{Eu}$ | 0.01 | 0.41 | 0.38 | 1.06 |
| $c^v_{Gd}$ | 0.00 | 0.00 | 0.00 | 3.28 |
| $c^v_{Tb}$ | 0.06 | 0.73 | | |
| $c^v_{Yb}$ | 0.10 | 1.14 | 0.75 | 2.14 |
| $c^v_{Lu}$ | 0.01 | 0.14 | | |

A more accurate Cl molality $m^m_{Cl}$ for the granitic melt may be obtained by using the methods proposed by [20], but their methods require the chemical composition data of apatite, or the bulk composition of aplite associated with the main granite intrusion. Here, using the value of $m^m_{Cl}$ obtained by Equation (1) and the Cl partition coefficient ($k^{Cl}_P$) [17,21,22], one can readily estimate the value of the Cl molality ($m^v_{Cl}$) of the magmatic fluid phase in equilibrium with the granitic melt at magmatic P–T conditions. As mentioned above, $k^{Cl}_P = m^v_{Cl}/m^m_{Cl}$, which is a function of pressure and independent of temperature within the continental crust [17,18]. Thus, $m^v_{Cl}$ can be obtained from Equation (2) below, if the value for $m^m_{Cl}$ is given (column A in Table 4).

$$m^v_{Cl} = k^{Cl}_P \times m^m_{Cl} \qquad (2)$$

Chlorine partition coefficients, $k_P^{Cl}$, between chloride-bearing fluids and granitic melts were determined experimentally over a pressure range of 2 to 8 kb and near the 700 and 750 °C isotherms by [21]. Their results indicate that the $k_P^{Cl}$ values are strongly dependent upon pressure. The values increase from 43.5 at 2 kb to 83.3 at 6 kb, and then decrease sharply to 13.0 at 8 kb. [22] reported a similar result for the pressure range of 2 to 4 kb, although a F-free system has higher $k_P^{Cl}$ values. These experimental data suggest that Cl prefers partitioning into aqueous fluids under magmatic conditions. However, experiments by [22] have also shown that Cl is compatible in granitic melts with a high F (>7 wt %) and low Cl (0.12 wt %) contents at 2 kb and 1000 °C.

Using the salinity data from fluid inclusions to estimate the Cl molality ($m_{Cl}^v$) is a quick solution for measuring the magmatic Cl content, unless Cl has been lost during cooling of the granitic melts. However, this only works if the trapped inclusions are primary magmatic fluids sourced from the granitic magma (intrusion) [6,10]. For example, the fluid inclusions trapped in acicular apatite enclosed in a plagioclase crystal from a granodiorite sample collected from the Lake George granodiorite stock represent typical magmatic fluids with a medium salinity of 15 wt % NaCl$_{eqv}$. This salinity is suitable for calculating the Cl molality ($m_{Cl}^v$) of the fluids [5,7] (column B in Table 4). Such a direct approach may avoid the complexity of estimating $m_{Cl}^m$ [20,24] and the use of Equation (2), which relies on the pressure-dependant $k_P^{Cl}$ parameter [21–23]. Medium to high salinity fluids (e.g., 17 wt %, NaCl$_{eqv}$., ~3.50 $M$ Cl$^-$), which are slightly lower salinity than the threshold value required for the maximum value of $k_P^{Yb}$ (see Table 3), are commonly associated with porphyry Cu–Au (–Mo) systems [26,27]. These values may be inputted into the equations to calculate $k_P^{REE}$ and then to calculate the REE concentrations of the magmatic fluids that are listed in Table 4, column D. High salinity fluids responsible for potassic alteration associated with porphyry Cu–Au (–Mo) deposits (52 wt % NaCl$_{eqv}$., 18.52 $M$ Cl$^-$) [27] have much higher values of $m_{Cl}^v$ than the threshold values (Table 3) required to reach the maximum values of $k_P^{REE}$ for all REE (Table 3; Figure 2). Thus, using the maximum $k_P^{REE}$ values (Table 3) to calculate REE contents in magmatic fluids is reasonable (see column E in Table 4). Such magmatic fluids of medium to high salinity leads to Cl molality values that are close to and higher than the threshold values. Thus, the calculated REE fluid–melt partition coefficients are relatively close to the maximum values.

## 2.3. Calculation of REE Concentration $C_{REE}^v$ in the Magmatic Fluid Phase

The REE concentration, $C_{REE}^v$, in the magmatic fluid phase can be readily calculated in terms of the $k_P^{REE}$ values, as described above, and the bulk-rock (melt) REE content $C_{REE}^m$. For example, Table 4 shows that the La content (3.05 ppm) in the magmatic fluid phase is computed by multiplying the value of $k_P^{REE}$ (0.099) with the La content (30.9 ppm) in the melt, which has been approximated from the analysis of the granite sample (column A in Table 4).

Utilizing this method, an Excel spreadsheet (see Supplementary Table S3) has been created that can be used to calculate $m_{Cl}^v$ from whole-rock Cl analytical data, and/or from the salinity data of fluid inclusions as well as to calculate $k_P^{REE}$ and REE concentrations in magmatic fluids associated with granitic melts.

## 3. Application

Two examples are presented here to show how to calculate REE concentrations in a magmatic fluid associated with (1) the Lake George granodiorite stock, which is thought to be genetically responsible for the formation of the Sb–Au–W–Mo mineral deposit, New Brunswick (Canada); and (2) the Bakircay Cu–Au (–Mo) porphyry system, which is in northern Turkey. The Lake George deposit was the largest antimony producer in North America until the mid-1990s. It is temporally and spatially associated with the Early Devonian Lake George granodiorite stock [28–34]. The styles of Au mineralization include Au-bearing quartz–carbonate veins, veinlets, and stockworks that are present within the granodiorite stock, quartz-feldspar dyke, and proximal metamorphic aureole, respectively; these are associated with earlier W–Mo mineralization. These characteristics suggest that the Lake George granodiorite intrusion

and related hydrothermal systems may have ultimately resulted in Au mineralization [28–31,33], resembling intrusion-related Au systems [7,34–36].

Table 4 lists the analyses of the Lake George and Bakircay granodiorites, Cl contents either from bulk analysis or from fluid inclusion salinity data, calculated Cl molality ($m_{Cl}^v$), calculated $k_P^{REE}$, and REE concentrations of magmatic fluids. Other data include the Lake George mineralized quartz-feldspar porphyry [33], the Bakircay potassic altered rock [4], and calculated REE contents in magmatic fluids with different $m_{Cl}^v$. These are tabulated in Supplementary Table S2.

The Lake George granodiorite-normalized REE distribution patterns for magmatic fluids (see Table 4 for the estimated value of $C_{REE}^v$ for each element) at a Cl molality ($m_{Cl}^v$) of 1.23 M and 3.02 M are plotted in Figure 4a, indicating that the REE concentrations in the magmatic fluids are remarkably elevated as the value of $m_{Cl}^v$ increases (Table 4). Also, the REE pattern of the fluid displays a pronounced negative Eu anomaly at $m_{Cl}^v$ equal to 1.23 M, whereas the Eu anomaly becomes less pronounced when $m_{Cl}^v$ equals 3.02 M. Figure 4b shows REE patterns in the calculated magmatic fluids at $m_{Cl}^v$ = 3.50 M and with the threshold values (Table 3) normalized by the Bakircay granodiorite. Although their LREEs are similar, the fluid with the threshold Cl⁻ molality displays relatively elevated HREEs compared to the granodiorite.

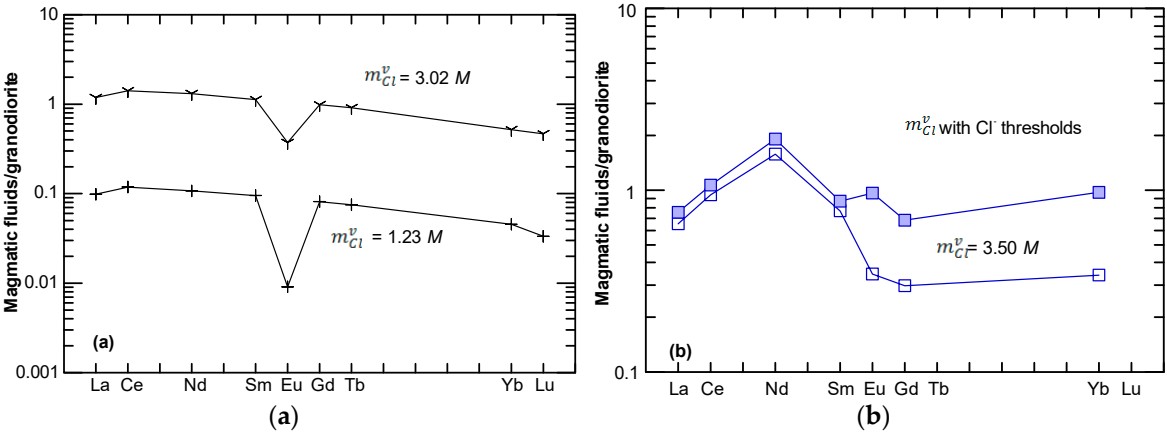

**Figure 4.** Lake George granodiorite-normalized REE distribution patterns of magmatic fluids with a Cl molality of $m_{Cl}^v$ equal to 1.23 to M and 3.02 M (**a**), and the Bakircay granodiorite-normalized REE distribution patterns of magmatic fluids with $m_{Cl}^v$ values equal to 3.50 *M* and the threshold value (**b**). The normalization values are taken from Table 4.

Figure 5a shows chondrite-normalized REE patterns for the Lake George granodiorite, mineralized (or altered) quartz-feldspar porphyry, and the calculated magmatic fluids, suggesting that these fluids are likely to decouple from the granodiorite, although the altered porphyry lacks a Eu anomaly. A higher $m_{Cl}^v$ fluid appears to be required to reduce the Eu anomaly, which is supported by fluid inclusion studies (i.e., halite-bearing inclusions) [7]. Such fluids with varied $m_{Cl}^v$ deriving from progressively cooling magmas would interact with the quartz-feldspar porphyry, resulting in the reduction to disappearance of the Eu anomaly and HREE abundances falling below that of the granodiorite. This process may have produced hydrothermal alteration and simultaneously Au mineralization in the vein stockwork systems [30,33], which is consistent with evidence from the lithogeochemistry, mineral chemistry, fluid inclusion data, and stable isotope data [7,34–37]. Notably, increasing the values of $m_{Cl}^v$ could raise the $k_P^{REE}$ values, and thus raise the REE concentration of the ore fluids resulting in LREE enrichment along with further reducing the Eu anomaly. This is also seen in magmatic fluids that are responsible for the origin of the intrusion-related Au–Sb deposit in northeastern Russia, which showed LREE enriched patterns, as reported in [10].

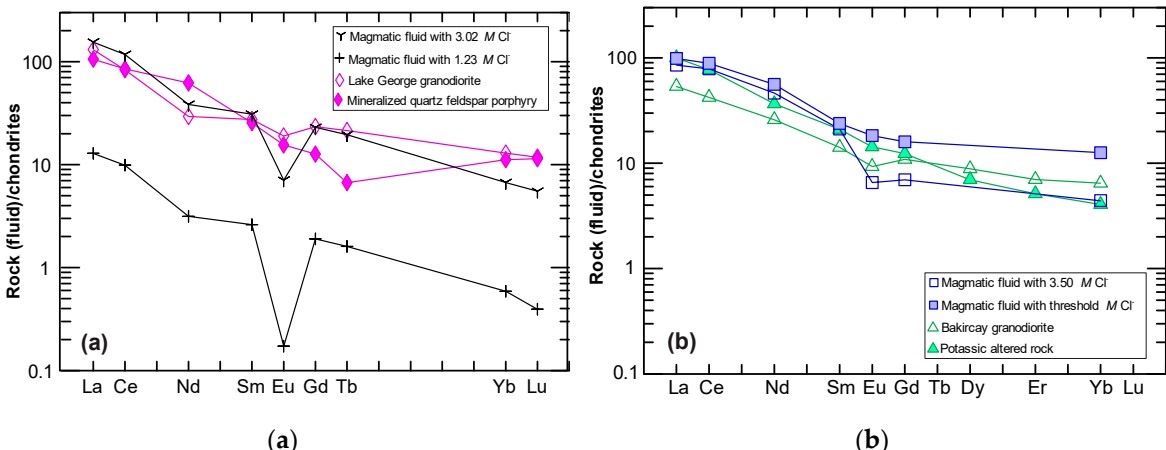

**Figure 5.** (**a**) A comparison of chondrite-normalized REE patterns of calculated magmatic fluids, the Lake George granodiorite and mineralized quartz-feldspar porphyry; (**b**) chondrite-normalized REE patterns of the Bakircay granodiorite, potassic altered rock, and the calculated magmatic fluids. The chondrite normalizing values are from [38].

Figure 5b shows the calculated magmatic fluids with an $m_{Cl}^v$ value of 3.50 *M* and with a threshold value associated with the Bakircay granodiorite melt. These are compared to potassic altered rock that hosts Cu–Mo mineralization [4], indicating that the mineralization is most likely to be related to ore fluids that have medium to very high salinities. The interaction of such fluids with the host rocks would have ultimately resulted in potassic alteration and ore mineral (e.g., chalcopyrite) precipitation. Such magmatic fluids with LREE, which are enriched relative to HREE and released from cooling magmas, are consistent with the mass-balance studies by [4] and numerical simulation, which are also based on REE fluid–granitic melt partitioning by [2].

To further test the validity of the method outlined in this study, a comparison has been made by using the REE data from natural fluid inclusions, which is analyzed by inductively coupled plasma mass spectrometry (ICP-MS) [6], with calculated magmatic fluids in equilibrium with a porphyritic granite, an aplite, and a granophyric granite from the Capitan Pluton, New Mexico, U.S.A. [39] (Figure 6; see Supplementary Table S2). Clearly, the chondrite-normalized REE patterns from both the fluid inclusions with high salinities (~80% NaCl$_{eqv.}$ [6], i.e., $m_{Cl}^v$ equals ~63.4 M, which is much higher than the threshold values; see Table 3) and the modeled magmatic fluids display many similarities. This includes LREE enriched relative to HREE and pronounced Eu negative anomalies (Figure 6). This comparison supports the conclusion that the fluid inclusions trapped in the quartz-vein system were derived from magmatic fluids, and that REE variation in the fluids reflects fractional crystallization of the Capitan pluton [6]. Note that the maximum values for $k_P^{REE}$ are used in the calculation of the REE concentrations in the magmatic fluids released at different stages of pluton evolution.

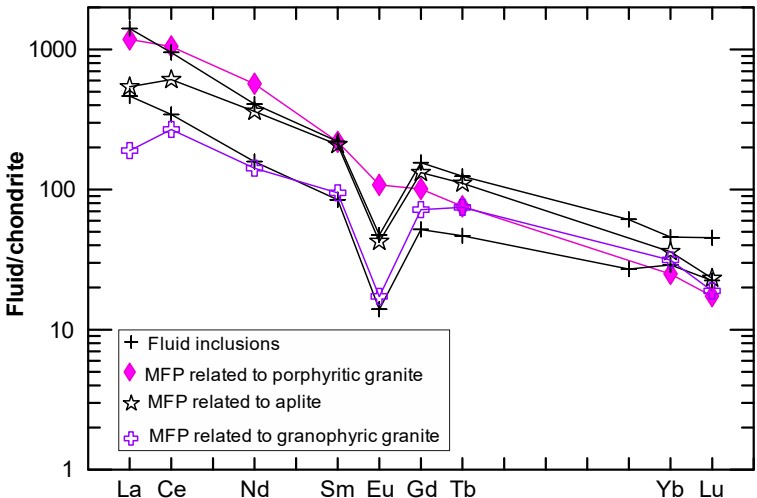

**Figure 6.** A comparison of chondrite-normalized REE patterns of the fluid inclusions (analyzed by inductively coupled plasma mass spectrometry, or ICP-MS) [6] with calculated magmatic fluid phases (MFP) related to the porphyritic granite (sample A29), the aplite (A17), and the granophyric granite (A12) from the Capitan Pluton, New Mexico, U.S.A. [39] (Supplementary Table S2). The chondrite normalizing values are taken from [38].

## 4. Discussion

A method is presented here that allows for the estimation of REE concentrations in magmatic fluids, which have separated from water-saturated granitic magmas. Then, these calculated REE concentrations are compared to those in intrusion-related hydrothermal alteration zones consisting of Au and porphyry Cu–Au (–Mo) deposits. This comparison helps to provide clues as to the origin of the ore fluids and metallogeny associated with deposits such as the Lake George Au–Sb deposit and the Bakircay Cu–Au (–Au) porphyry system. A number of studies have shown that the Lake George intrusion-related Au–Sb deposit may have been dominantly formed by a magmatic fluid phase exsolved from a granodiorite intrusion. This conclusion is based on evidence from the field relations, petrology [30], geochemistry [31], stable isotopes (O, H, S), and microthermometry [7,28,29,33,34]. The Bakircay Cu–Au (–Mo) porphyry system is thought to have been the product of magmatic fluids in terms of the style of its alteration, its lithogeochemistry, and nature of the fluids [2,5].

The key is to modeling these two systems is to estimate the empirical values of $k_P^{REE}$, which can be calculated using the polynomial equations in Table 2. These equations are based on a statistical analysis of the existing experimental dataset (Supplementary Table S1) and indicate that the values of $k_P^{REE}$ (Figure 1 and Table 2) achieve a maximum when the Cl molality ($m_{Cl}^v$) values are equal to the threshold values outlined in Table 3. These threshold values provide a theoretical constraint on the solubility of REE in magmatic fluids containing chloride. When $m_{Cl}^v$ is higher than these threshold values, the REE chloride complexes (e.g., $LaCl_3$, $EuCl_2^-$, $EuCl_3$, $YbCl_3$) become unstable under the magmatic hydrothermal conditions, allowing for the potential precipitation of REE-bearing minerals. However, the predictions outlined by these threshold values for the Cl molality needs to be further tested experimentally using high-salinity solutions.

The experimental data used for this study only considers Cl [1–3] and has not considered other chemical components such as $CO_3^{2-}$ or $HS^-$ and their probable effect on REE fluid–melt partition coefficients ($k_P^{REE}$). Despite the difficulty of determining the Cl molality ($m_{Cl}^v$) of a granitic melt [24], this study uses bulk-rock analysis to estimate the minimum value of $m_{Cl}^v$ in the granitic magma (melt), although using salinity data from primary fluid inclusions may be a quicker solution to this problem (Table 4; Supplementary Table S3).

The values of $k_P^{REE}$ in this study are only related to $m_{Cl}^v$ (Table 2); this value is linked to $k_P^{Cl}$, which is mainly pressure dependent [21]. Therefore, the equations proposed in this study may be applied

to a wide pressure range within the continental crust, given that a proper value of $k_P^{Cl}$ at a specific pressure [21–23] is used to calculate the value of $m_{Cl}^v$. As pointed out above, this problem may be readily resolved by using magmatic fluid inclusion data to estimate $m_{Cl}^v$ (Table 4). When the values of $m_{Cl}^v$ reach their threshold values (Table 3), $k_P^{REE}$ reach their maximum values, and therefore must be independent of pressure, as indicated by experimental work [3].

Furthermore, LREE and HREE display a distinct behavior in magmatic fluids associated with granitic magmas, leading to their fractionation in magmatic hydrothermal systems. The examples given by this study confirm that the calculated magmatic fluids are enriched in LREE relative to HREE (Figures 5 and 6), and that Eu appears to deviate from the other REEs with either having pronounced negative anomalies in low to medium $m_{Cl}^v$ fluids or without notable Eu anomalies in high $m_{Cl}^v$ fluids.

In these calculations and estimations, the oxygen fugacity is not considered, although it is known that the Lake George granodiorite exhibits characteristics of a reduced I-type [34–36], whereas the Bakircay granodiorite is a normal oxidized I-type based on its mineral assemblage [4]. This suggests that the behavior of Eu in the magmatic hydrothermal systems could also have been influenced by redox conditions [1]. The Lake George granodiorite contains primary magmatic pyrrhotite and ilmenite, but lacks magnetite [28–30]. The pyrrhotite would be unstable when interacting with Cl-bearing magmatic fluids, as shown by [35,40–43]. This would help to liberate Au and S into the ore fluids [35,40–44], and thus allowed for the precipitation of ore minerals including Au in shear zones and hydrofractures [7,34–36,44]. Some primary magmatic sulfides contain remarkably high amounts of Au (up to 20.7 ppm), as determined by laser ablation ICP-MS [35], suggesting that the decomposition of these sulfides, as a result of Cl-bearing magmatic fluids, may have provided an important mechanism for intrusion-related Au mineralization [43,44] and porphyry Cu–Au systems [40–42].

Although intrusion-related Au systems such as the Lake George Sb–Au–W–Mo mineral deposit are commonly characterized by low to medium salinity, carbonic, and reduced ore fluids [7,43], the presence of high-salinity fluids appears to be required to balance the LREE abundances, the Eu anomaly, and HREE abundances, as observed in the mineralized porphyry [33] and by the modeling presented in this study. Such a high-salinity fluid is evident by the occurrence of halite-bearing fluid inclusions [7,28,29]. Although $CO_2$-rich fluids with a much lower salinity are evident in late fluid inclusion assemblages [7,28,29], they are unlikely to impact on the REE partitioning between earlier magmatic fluid and granitic melts. On the other hand, porphyry Cu–Au (–Mo) systems are characterized by oxidized, medium to high-salinity (or $m_{Cl}^v$) ore fluids [26,27] associated with the granitic intrusions. Such conditions are favorable for promoting Cu enrichment during magmatic hydrothermal evolution [17–21]. It is worth noting that the calculated REE patterns of moderate to high-salinity magmatic fluids and altered rocks display a notable tetrad effect for the lanthanides, especially for the LREE (Figure 5a,b and Figure 6). This tetrad effect may possibly be a result of the interaction of the magmatic fluids with the surrounding country rocks (cf. [45]).

## 5. Conclusions

LREEs behave differently in magmatic fluids associated with granitic magmas. They are either fluid compatible in higher $m_{Cl}^v$ magmatic fluids or low $m_{Cl}^v$ granitic melts, whereas HREE are exclusively magmatic fluid incompatible, preferring the granitic melt. Consequently, magmatic fluids tend to be rich in LREE relative to HREE, resulting in REE fractionation during the evolution of magmatic hydrothermal systems. When the value of $(m_{Cl}^v)^3$ reaches a threshold value, REE fluid–granitic melt partition coefficients $k_P^{REE}$ achieve their respective maximum values, suggesting that magmatic fluids associated with granitic magmas will not dissolve any more REEs than the predicted maximum values. Europium behaves differently from the other REEs, requiring much higher $m_{Cl}^v$ values to become fluid compatible, and thus magmatic fluids with low $m_{Cl}^v$ will have a negative Eu anomaly.

REE concentrations in magmatic fluids associated with granitic melts (intrusions) may be estimated in terms of the polynomial equations developed in this study (Table 2), and then applied to altered and/or mineralized rocks to model the origin of ore fluids. This technique has been applied to the

Lake George Sb–Au–W–Mo mineral deposit, New Brunswick, Canada, and Bakircay Cu–Au (–Mo) porphyry systems in northern Turkey. Modeling of either system suggests that the ore fluids may have been dominated by magmatic fluids, albeit with different Cl concentrations, i.e., the former with low to medium values of $m_{Cl}^v$, and the latter with medium to high values of $m_{Cl}^v$.

**Supplementary Materials:** The following are available online at http://www.mdpi.com/2075-163X/9/7/426/s1, Table S1. Experimental REE fluid–granitic melt partition coefficients and chlorine molalities (unit in M) of fluids in equilibrium with granitic melts. Table S2. REE concentrations (ppm) in rocks, calculated magmatic fluids, and natural fluid inclusions. Table S3. Procedures of calculating the Cl molality of magmatic fluids from whole-rock analysis and salinity of fluid inclusion as well as REE fluid–melt partition coefficient and REE concentrations in magmatic fluid (this is a digital version of Table 4, containing formulas).

**Funding:** This study did not receive any specific grant from funding agencies in the public, commercial, or not-for-profit sectors.

**Acknowledgments:** The constructive review of an earlier draft of manuscript by Sean H. McClenaghan is gratefully acknowledged, which greatly improved the manuscript. Discussion with Simon Gagne is appreciated. I thank three anonymous journal reviewers for their constructive comments on the manuscript, which significantly improved the presentation of this study. The Academic Editor of the journal is gratefully acknowledged for handling and technically editing the manuscript and for encouraging me to resubmit this paper.

**Conflicts of Interest:** The author declares no conflict of interest.

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
