# Peer review of "Using Rare Earth Elements (REE) to Decipher the Origin of Ore Fluids Associated with Granite Intrusions"

_minerals, doi:10.3390/min9070426_

Reviewer 1 Report

Below are my comments on the authors response to my previous comments:

Reply: Thanks for the comment and detailed editing the manuscript (MS). Based on these, I have revised MS accordingly.

Apart from the corrections to the grammar and the addition of a few extra sentences this manuscript remains essentially unchanged and hence, there is little reason for me to change my opinion.

Reply: The major critique by Reviewer 1 is that the datasets used for statistical analysis by this

study have a large gap between the low and high chlorine molality as shown in Figure 1. These

data are cited from references [1-3] that were acquired by their high P-T experiments. I have

noticed that there is a linear relationship present between the REE fluid-melt partition

coefficients and chlorine molality of the fluids that have low chlorine molality [5], but such a

relationship cannot be applied to the fluids with high chlorine molality. Therefore, this study

has tried to best fit the available dataset statistically as shown in Figure 1, and obtained to a set

of new polynomial equations that may be used for estimating REE fluid-melt partition

coefficients. As pointed out by Reviewer 1, there is a data gap as shown in Figure

1. This problem cannot be resolved until more new high P-T experiment work is conducted to

fill the data gap. I stress that the predicted trend and the relationships presented in this study

need to be tested by future experimental work to verify if they are correct. More importantly,

this study shows the clues of how to use REE to study the origin of ore-fluids associated with

granite intrusions..

The author admits that the data gap creates a problem for the curve fitting but has not changed the approach to the problem. I still find the statement that “These equations may be applied to the entire pressure range (0.1 to 10.0 kb) within the continental crust” of great concern. As I said before, I don’t think it is valid to extend the curve fitting beyond the salinity ranges constrained by the experimental data, especially for two such disparate datasets.

Reply: Reviewer 1 would like see REE analytical data of fluid inclusions, which is again

beyond the objective of this study. I have used the published data from two examples to show

REE characteristics of the calculated magmatic fluids and compared them with those of altered rocks related to mineralization, suggesting that the magmatic fluids may have played a role in the formation of ore deposits.

The objective of this study was to use the new polynomial equations to track the sources of ore-fluids responsible for the formation of presumably granite-related mineral deposits. I do not see how including some published data for REE in fluid inclusions is beyond the objectives of this study and I highly recommend that you do so.

Reply: As mentioned above, the experimental and analytical data used in this study are

all cited from references. Based on a statistical analysis of the available experimental data,

a new set of equations are obtained for estimating REE fluid-granitic melt partition

coefficients; Then, this study uses two publishing cases as an example to show how to

apply the estimated partition coefficients to calculate REE contents in magmatic fluids

associated with granitic melts.

Thus, I am surprised that the author has chosen two mineral deposits with relatively little available data! Including some REE data from fluid inclusions should be part of the objectives of this study. I suggest that the author include some calculations based on data from the following deposits in the examples as they contain actual data on REE in the fluids:

Banks et al. (1994) REE composition of an aqueous magmatic fluid: A fluid inclusion study from the Capitan Pluton, New Mexico, U.S.A., Chem. Geol. Volume 113, Issues 3–4, Pages 259-272.

Vikent’eva et al. (2012) REE in fluid inclusions of quartz from gold deposits of north-eastern Russia. Central European Journal of Geosciences, Volume 4, Issue 2, pp 310–323

Pandur et al. (2014) HYDROTHERMAL EVOLUTION IN THE HOIDAS LAKE VEIN-TYPE REE DEPOSIT, SASKATCHEWAN, CANADA: CONSTRAINTS FROM FLUID INCLUSION

MICROTHERMOMETRY AND EVAPORATE MOUND ANALYSIS. The Canadian Mineralogist Vol. 52, pp. 717-744

Zhang & Audetat (2017) What Caused the Formation of the Giant Bingham Canyon PorphyryCu-Mo-Au Deposit? Insights from Melt Inclusions and Magmatic Sulfides. Economic Geology, Vol. 112, pp. 221–244.

Other Comments:

Line 8. I don’t like the use of the term Magmatic Vapor Phase. As discussed in Cline & Bodnar JGR (1991) the fluid that evolves from a magma may be a high density liquid, a low density vapor or a mixture of vapor + liquid (immiscible fluid). By denoting the fluid as a vapour you are invoking a phase which typically cannot contain the high concentrations of Cl as used in your calculations.

Lines 226 – 239. I note that you have used the approx. maximum salinity of fluid inclusions from the Lake George granodiorite although your 2004 paper states they range from 0 to 15 wt.% NaCl (quite a large range!). However, for the Bakircay Cu-Au (-Mo) porphyry system you have taken a generic value from the literature which may not be at all related to the fluids at this deposit. Note that Cline & Bodnar JGR (1991) state that the salinity of porphyry systems may vary between 2 and 84 Wt.% NaCl. If you are using fluid-inclusions to estimate the chlorine molality as stated in the text then you should choose a deposit where the inclusion salinity is known.

Line 277. In Figure 4 why do you calculate the REE patterns in Fig 4a using a 3.02 M fluid but in Fig 4b you use a 3.5 M fluid? It would be far better for comparison if the calculations used the same salinity fluid.

Line 303. In Figure 5a the calculated values for the porphyry deposit are in reasonable agreement with the calculated values for the 3.02 M fluid but the calculated values of the Lake George deposit are a poor match and an order of magnitude higher than the calculated values for the 1.23 M fluid. This suggests that the alteration is not related to the reported chemistry of the fluid inclusions at the Lake George deposit.

Therefore, I recommend that the manuscript is rejected. It requires more experimental and analytical data to validate the assumptions made in the paper.

Author Response

Please find attached cover letter.

Reviewer 2 Report

The reviewer thinks that the manuscript was revised based on the previous comments. Therefore, I recommend to be accepted in Minerals.

Author Response

Reply: I thank Reviewer 2 for the recommendation of accepting this manuscript

Reviewer 3 Report

Manuscript ID: minerals-529023

Using rare earth elements (REE) to study the origin of ore-fluids associated with granite intrusions

Yang focuses on a practical method to estimate REE concentrations in magmatic vapor phase in equilibrium with a granite melt from the estimation of fluid-melt partition coefficients of REE, which depends on the chlorine concentration. This methodology has been previously addressed by the author in another journal (The Open Geology Journal, 2012, 6, 19–24), but it was only usable for a molality ≤ 3.5. This new contribution tries to be an extension able to estimate the fluid/melt partition coefficient of REE for higher molality values (>3.5M). New polynomial equations have been achieved by the author, obtaining a good match with available experimental data and most of REE. According to the experimental work on which this approach is based, the latter would not depend on temperature and would be applicable for a wide range of pressures (0.1 to 10 kb) that well cover the pressures of granite emplacement in the crust. The author finds an important point with the new extended formulation: the existence of a threshold above which, as the molality of chlorine increases, the distribution coefficient of the REEs hardly changes. Although more experimental work would be necessary to confirm this, as there is an important experimental gap, especially concerning chlorine concentration, this finding itself is a good contribution by the author. The existence of this threshold also allows the author to establish that the HREE will always be compatible in the melt, which is also an important finding on the part of the author. In contrast, LREE behavior will depend on the chlorine concentration, remaining in the melt at low chlorine concentration but being exclusively incompatible in the melt at high chlorine contents. As a result, the author concludes that there is REE fractionation during the evolution of magmatic hydrothermal systems. The author uses two real examples in order to show the use of the method. Both examples are based on host granodiorites exhibiting a strong metasomatism with different ore, and where, a priori, the exsolved magmatic fluid has contrasted chlorine contents. 

The study is well presented and written though clarity can be improved. I agree with most of conclusions, but I have some concerns with some aspects, and some statements 

I recommend it for publication after minor corrections.

The considerations to improve the manuscript are presented below and in the attached PDF version.

Author Response

Thirdly, I reply to the Comments by Reviewer 3.

Yang focuses on a practical method to estimate REE concentrations in magmatic vapor phase in equilibrium with a granite melt from the estimation of fluid-melt partition coefficients of REE, which depends on the chlorine concentration. This methodology has been previously addressed by the author in another journal (The Open Geology Journal, 2012, 6, 19–24), but it was only usable for a molality ≤ 3.5. This new contribution tries to be an extension able to estimate the fluid/melt partition coefficient of REE for higher molality values (>3.5M). New polynomial equations have been achieved by the author, obtaining a good match with available experimental data and most of REE. According to the experimental work on which this approach is based, the latter would not depend on temperature and would be applicable for a wide range of pressures (0.1 to 10 kb) that well cover the pressures of granite emplacement in the crust. The author finds an important point with the new extended formulation: the existence of a threshold above which, as the molality of chlorine increases, the distribution coefficient of the REEs hardly changes. Although more experimental work would be necessary to confirm this, as there is an important experimental gap, especially concerning chlorine concentration, this finding itself is a good contribution by the author. The existence of this threshold also allows the author to establish that the HREE will always be compatible in the melt, which is also an important finding on the part of the author. In contrast, LREE behavior will depend on the chlorine concentration, remaining in the melt at low chlorine concentration but being exclusively incompatible in the melt at high chlorine contents. As a result, the author concludes that there is REE fractionation during the evolution of magmatic hydrothermal systems. The author uses two real examples in order to show the use of the method. Both examples are based on host granodiorites exhibiting a strong metasomatism with different ore, and where, a priori, the exsolved magmatic fluid has contrasted chlorine contents. The study is well presented and written though clarity can be improved. I agree with most of conclusions, but I have some concerns with some aspects, and some statements 

I recommend it for publication after minor corrections.

The considerations to improve the manuscript are presented below and in the attached PDF version.

Reply: I highly appreciated for the recommendation of Reviewer 3 who have pointed out the value of this study and made detailed editing of the MS that helps a lot for doing this revision.

1) The R2 parameter in equations for estimating the partition coefficient fluid/melt in REEs. The method used by the author to estimate the match between chlorine molality and fluid/melt partition coefficient is parameter R2. The author calls it the correlation coefficient, but in reality, the correlation coefficient is only R. The coefficient R2 is called the determination coefficient and the author should correct it in the text.

Reply: This is fixed in the revised MS.

2) The section 2.2. Calculation of chlorine molality. In the Bakircay granodiorite example, the author uses the salinity of fluid inclusions of rocks strongly metasomatized to estimate the molality of the fluid, but this is a bit risky, as the fluid may have been contaminated with resident fluids from the wall rock, leading to a change in its composition. The author must give arguments to rule out a possible contamination of this fluid with the resident fluids.

Reply: Agree. Because the fluids responsible for strongly metamomatized (altered and mineralized) rock at Bakircay are those with medium to high salinity, the chlorine molality values are close to and higher than the threshold. In other words, the calculated REE fluid-melt partition coefficients are pretty close to the maximum values, which could approximately avoid the effect of the resident fluids that must have reduced the chlorine molality, given potential contamination took place.

3) Discussion section. The author needs deal with certain points and the section must be enlarged.

a. The author must give arguments so that the reader has no doubt that the two cases evaluated are really deposits related to a fluid phase separated from the melt, and not a late hydrothermal event disconnected of the magmatic activity. There are plenty of good examples where gold deposits

are disconnected of the magmatic activity, for example in Permian gold deposits from Europe.

Reply: Thanks for the suggestion. The data from field relations, petrography, stable (O, H, S), lithogeochemistry, microthermometry, and alteration style point to that both the Lake Gorge and the Bakircay deposits are genetically associated with magmatic fluids exsolved from their respective granodiorite intrusions. The other case (e.g., the Capitan Pluton, New Mexico, USA ) is cited in the revised MS to indicate the origin of extremely high salinity fluids are derived from the magmas that had undergone fractional crystallization [6, 39].

b. The author should also give some brushstrokes about how the separation of volatiles occurred in both deposits: e.g., second boiling, low emplacement pressure, strong undercooling, etc. P-T emplacement

conditions could be a useful tool for it.

Reply: This issue was discussed in detailed for the Lake George deposit by Yang and Lentz ([37]) using petrological data and mineral chemistry data of the rock-forming minerals and related geothermobarometry. Less work was done at the the Bakircay deposit, although evidence from petrography, lithogeochemistry and alteration zonation suggest that the separation of volatiles took place at shallow depth [4].

c. The author should consider that whole-rock REE analyses of the rock with strong potassium alteration from the Bakircay granodiorite, and probable the same in the mineralized quartz feldspar porphyry from the Lake George granodiorite; could be the result of REE partitioned into the fluid separated from the melt, but also of REE from the leaching of REE carrying minerals from the wallrock, e.g. apatite, monazite etc. Leaching of both minerals has been frequently reported in strong metasomatized rocks like episyenites. The author should address and discuss this possibility.

Reply: The whole-rock REE concentrations of potassium alteration from the Bakircay granodiorite and from the mineralized quartz feldspar porphyry at Lake George are presented in Supplementary Table 2, which show similarities to the calculated magmatic fluids associated with the granodiorite intrusions, respectively. So far, there is a lack of evidence that REE leached from REE-bearing minerals in the wallrock to involve in magmatic fluids considered, so that such a possibility is not further discussed in this study.

d. At the beginning of the Discussion section the author makes a brief introduction about the experimental data on which the formulations obtained are based do not consider components such as CO32- and HS- and the fact of REEs prefers entering carbonate melt. A priori, this could be a serious inconvenience, since, if the fluid separated from the melt, as the author seems to be saying from fluid inclusions, corresponds to a CO2-rich fluid, the cases chosen as real examples might not be the best. However, the REE patterns in figure 5, especially those of the Bakircay fit quite well to the metasomatized rock. Thus, the author should allude to this good fit and say that it does not seem that the CO2 content has had a special relevance in the final result.

Reply: Thanks for the good points. I have moved the beginning part of the Discussion section down to the second paragraph as suggested by Reviewer 3. The modelling indicates that the moderate salinity fluid trapped in acicular apatite enclosed by plagioclase crystals [7] resembles the mineralized porphyry in REE patterns, but low salinity CO2-bering fluid exhibits a remarkable difference. This latter fluid is likely to reflect the mixture of magmatic fluid and circulated meteoric water late evidenced by O, H, and S isotopes [7, 28-29]. 

e. The author's study theme, with REE and a fluid phase that interacts with rocks is the ideal setting for the tetrad effect of lanthanides. The author should discuss and give arguments if his samples present this effect and explain what effects the lanthanide fractionation could have in its formulation, even briefly.

Reply: Agee. This tetrad effect of lanthanides due to fluid and rock interaction is briefly discussed.

4) Supplementary data.

One way to increase the diffusion of this great work is to supply in the supplementary

material an Excel sheet that calculates the DREEfluid/meltand molality from fluid inclusions

data. I think the scientific community will appreciate it.

Reply: Thanks for this great idea. A supplementary Table 3 (Excel sheet) is added, which can be used for calculating REE fluid-melt partition coefficients and chlorine molality from fluid inclusion data.

SPECIFIC COMMENTS (see below)

Reply: All suggestions in the edited PDF MS are considered and corrected in this revision.

Round  2

Reviewer 1 Report

The author has used an empirical fitting method to join two disparate REE datasets and states that this fit can be used for most conditions encountered in the Earth's crust. However, I am still of the opinion that these datasets cannot be joined at present because more experimental data is required to fill in the large gap between high and low salinity experiments.The exact point where the so-called "threshold values" become applicable is still not defined.

The addition of an extra example which compares the calculated REE values with those in fluid inclusions is most welcome. However, the result is not surprising as the partition values are derived from the upper end (i.e. the threshold values) which are well constrained by experimental data.The validity of calculations relating to the region of no experimental data still remains to be tested. 

I have attached a copy of the manuscript with some minor corrections to the latest version.

Author Response

Reply: Thanks for reviewing the ms again. The suggested corrections by Reviewer 1 have been made or considered. Based on the experimental data available in the cited references, this study proposes a method to estimate REE fluid-melt partition coefficients over low to high chlorine molalities and to test it in natural ore systems. The results are encouraging.

Reviewer 3 Report

The author has addressed the considerations made in my previous review of the draft which makes the work better. I have only found two typos that I have reflected in the PDF attached. 

Author Response

Reply: Thanks for reviewing again. The two typos are corrected.